# Ceftazidime-Avibactam as Osteomyelitis Therapy: A Miniseries and Review of the Literature

**DOI:** 10.3390/antibiotics12081328

**Published:** 2023-08-17

**Authors:** Alessandro Mancuso, Luca Pipitò, Raffaella Rubino, Salvatore Antonino Distefano, Donatella Mangione, Antonio Cascio

**Affiliations:** 1Department of Health Promotion, Mother and Child Care, Internal Medicine and Medical Specialties “G. D′Alessandro”, University of Palermo, 90127 Palermo, Italy; 2Infectious and Tropical Disease Unit, Sicilian Regional Reference Centre for the Fight against AIDS, AOU Policlinico “P. Giaccone”, 90127 Palermo, Italy; 3Antimicrobial Stewardship Team, AOU Policlinico “P. Giaccone”, 90127 Palermo, Italy; 4Microbiology and Virology Unit, AOU Policlinico “P. Giaccone”, 90127 Palermo, Italy

**Keywords:** ceftazidime–avibactam, osteomyelitis, arthritis, *Pseudomonas aeruginosa*

## Abstract

Bone and joint infections (BJIs) caused by multidrug-resistant gram-negative bacteria are becoming a concern due to limited therapeutic options. Although not approved for these indications, an ever-growing amount of evidence supports the efficacy and safety of ceftazidime–avibactam as a therapy for osteomyelitis and prosthetic joint infections. Here, we present three cases of difficult-to-treat resistant *Pseudomonas aeruginosa* osteomyelitis that were successfully treated with ceftazidime–avibactam alone or in combination therapy with fosfomycin and amikacin. Ceftazidime–avibactam was prescribed at a daily dose of 2.5 g every 8 h for 42 days in all cases. One potential drug-related adverse effect was observed, i.e., *Clostridioides difficile* infection, which occurred after fourteen days of treatment with ceftazidime–avibactam.

## 1. Introduction

Bone and joint infections (BJIs) represent a significant therapeutic challenge and are associated with a high rate of relapse despite apparent treatment success. Gram-positive bacteria, particularly *Staphylococcus aureus*, are the most commonly isolated etiologic agents of BJI, with methicillin-resistant *S. aureus* responsible for some of these cases [1]. Although multidrug-resistant (MDR) and extensively drug-resistant (XDR) gram-negative bacteria (GNB) are rarely involved in osteomyelitis (ca. 10% of cases) [2], their prevalence has been increasing in recent years, highlighting the urgent need for new antimicrobial agents [3].

Ceftazidime–avibactam (CAZ–AVI) is a novel beta-lactam/beta-lactamase inhibitor containing a third-generation cephalosporin and avibactam, a non-beta-lactam beta-lactamase inhibitor [4]. The addition of avibactam to ceftazidime extends the spectrum of ceftazidime to include Enterobacteriaceae that produce ESBL, AmpC, KPC, and some class D serine β-lactamases, but not metallo-β-lactamase (MBLs). Ceftazidime–avibactam has variable activity against non-fermenting gram-negative bacteria like *Pseudomonas aeruginosa*. As long as resistance to antipseudomonal cephalosporins in *P. aeruginosa* is caused by the overproduction of an AmpC enzyme or, less frequently, by the acquisition of a class A β-lactamase, the addition of avibactam can restore ceftazidime’s activity against these resistant strains. Nevertheless, the available data suggest that avibactam does not recover the activity of ceftazidime as consistently as it does for Enterobacteriaceae, most likely due to the presence of additional resistance mechanisms, such as porin alterations, efflux pumps, metallo-β-lactamases, or OXA-β-lactamases [5,6].

The combination ceftazidime–avibactam is approved by the US Food and Drug Administration (FDA) and the European Medicines Agency (EMA) for infections without additional therapeutic options, including complicated intra-abdominal infections, complicated urinary tract infections, and hospital-acquired pneumonia/ventilator-associated pneumonia. BJI is not yet approved as an indication for ceftazidime–avibactam administration, and optimal treatment dosages and durations for osteomyelitis or arthritis have not been investigated.

In this paper, we describe three patients with difficult-to-treat resistant (DTR) *P. aeruginosa* osteomyelitis who were successfully treated with ceftazidime–avibactam and include a review of the literature on this topic.

## 2. Cases Description

### 2.1. Patient 1

An 89-year-old Caucasian man was referred to our hospital for treatment of calcaneal osteomyelitis; he had a previous history of a calcaneal burn ulcer, which occurred about 30 years ago, complicated by recurrent episodes of wound drainage. Past medical history included type 1 diabetes mellitus with poor glycaemic control (HbA1c = 7.8%) and complicated by diabetic retinopathy, arterial hypertension, and benign prostatic hyperplasia. Four months before hospitalization, due to the worsening of the calcaneal lesion, he was admitted to a clinic where he practised targeted therapy based on the isolation of *P. aeruginosa* from tissue sampling from the wound. He was admitted to the hospital once more three months later as a result of the return of pain, swelling, and secretion from the calcaneal lesion. Magnetic resonance imaging (MRI) was suggestive of osteomyelitis of the calcaneum. This time, he performed bone biopsies with isolation of *P. aeruginosa*.

Together with those of the other two patients described later, this strain of *P. aeruginosa* is reported in Table 1 with an antibiotic sensitivity profile (according to EUCAST clinical breakpoints v 13.1).

On admission, the patient was afebrile (36.6 °C) with a heart rate of 77 beats per minute and blood pressure of 120/70 mmHg. Physical exam was notable for a secretory calcaneal ulcer (stage IV, according to European Pressure Ulcer Advisory Panel [7]).

Laboratory examination was pertinent for a white blood cell (WBC) count of 6.4 × 10^3^ cells/μL (reference range 4–11 × 10^3^ cells/μL), blood urea nitrogen (BUN) 50 mg/dL (reference range 10–70 mg/dL), serum creatinine 0.81 mg/dL (reference range 0.67–1.17 mg/dL) and C-reactive protein (CRP) 10.57 mg/L (reference range <5 mg/L). Table 2 shows the laboratory examinations performed at admission and those performed on the other two patients described later.

On hospital day 1, based on the previous microbiological isolate from a bone biopsy, antibiotic therapy with ceftazidime–avibactam 2.5 g IV every 8 h was administered. After 14 days of antibiotic therapy, a sudden worsening of the clinical conditions with the appearance of abdominal pain and distension, fever, leucocytosis, and a rise in CRP (146.8 mg/L) and procalcitonin (4.8 μg/L, reference range <0.5 μg/L) occurred. Computed tomography (CT) of the abdomen showed marked oedema and inflammation of the rectum. In stool samples, an enzyme immunoassay detected *Clostridioides difficile* and toxin A/B. As a result, the patient underwent a 10-day course of fidaxomicin 100 mg OS every 12 h and metronidazole 500 mg IV every 8 h, with rapid clinical improvement. Ceftazidime–avibactam was continued.

The patient was discharged on the twenty-fifth hospital day with intravenous ceftazidime–avibactam for another two weeks to complete a course of six-week therapy. By the end of treatment, the patient had no signs or symptoms of infection, and the MRI showed no new bony destruction and resolution of soft tissue swelling. At the two-month follow-up visit, no recurrence of *C. difficile* infection was reported.

### 2.2. Patient 2

A 57-year-old Caucasian man with uncontrolled type 2 diabetes, rheumatoid arthritis, and a history of right trans-tibial amputation in September 2022 presented to our clinic in March 2023 with increased pain, purulent drainage, and erythema around the tibial stump. These symptoms had worsened the previous week but were unrelated to systemic symptoms.

On admission, a physical examination of the right tibial stump revealed a large area of substance loss with deep tissue exposure. The CT of the right tibial stump revealed a disrupted cortex, lytic lesions of the femoral condyles and at the proximal epiphysis of the right tibia, and soft tissue swelling, all suggesting osteomyelitis (Figure 1). On the seventh day, the patient underwent surgical debridement of the right tibial stump and soft tissue, and empirical antibiotic therapy was then initiated with the administration of cefepime 2 g IV every eight hours and fosfomycin 4 g IV every six hours. *Corynebacterium striatum* and *P. aeruginosa* were both detected in the bone biopsy culture, with the latter resistant to all beta-lactams examined (aside from ceftazidime–avibactam), fluoroquinolones, and aminoglycosides (Table 1). Based on the evidence of the microbiological isolate, cefepime was replaced by ceftazidime–avibactam 2.5 g IV every eight hours, fosfomycin was continued, and linezolid was added for the *C. striatum*.

After a six-week course of therapy, the patient was discharged home with significant clinical improvement. There was no sign of an infection recurrence at the two-month follow-up visit.

### 2.3. Patient 3

A 17-year-old Gambian man was admitted to Lampedusa’s health centre in April 2023 with trophic lesions on both feet. He had been walking through the Sahara Desert barefoot for two months. A CT scan that was performed in the emergency room revealed osteo-rarefaction of the right foot’s tarsus and metatarsal bones, and the patient was then transferred to our department for proper care due to the osteomyelitis diagnosis (Figure 2).

On admission, the patient was afebrile (36.7 °C), and laboratory examination showed a white blood cell count of 5.7 × 10^3^ cells/μL and C-reactive protein 0.6 mg/L. On the tenth day, the incision and drainage of the plantar portion of the distal and proximal phalanx of the big toe were conducted. As a result of the intraoperative cultures, revealing *P. aeruginosa* (Table 1), ceftazidime–avibactam 2.5 g IV every eight hours and amikacin 15 mg/Kg IV was administered, the latter for a total of two weeks.

The therapy was continued for a total of six weeks. The patient improved clinically without toxicity data. No additional surgical cleaning was necessary.

## 3. Discussion and Review of the Literature

These cases highlighted the few available therapeutic options for treating BIJs brought on by MDR/XDR-GNB and the need for new antimicrobial drugs.

In this series, all BJIs involved DTR-*P. aeruginosa* which was consistently resistant to ceftolozane–tazobactam, leaving ceftazidime–avibactam as the only viable therapeutic option. Clinical data on CAZ–AVI’s efficacy in treating BJIs are limited. Nonetheless, several reports mentioned that patients with osteomyelitis or arthritis caused by MDR/XDR-GNB were successfully treated with CAZ–AVI [8,9,10,11,12,13,14,15,16,17,18,19,20,21].

Considering the novelty of the research, we conducted literature research on PubMed and sites where the works submitted to the journals and not yet accepted appear. In detail, the websites listed in the following link https://asapbio.org/preprint-servers (accessed on 22 July 2023) were considered. The terms “avibactam” AND (“bone” OR “joint” OR “osteomyelitis” OR arthritis”) were used, and only papers reporting patients with BJIs treated with ceftazidime–avibactam were selected.

Thirteen case reports that described fourteen patients with BJI treated with CAZ–AVI which were published between 2017 and 2022 were selected for the analysis [8,9,10,11,12,13,14,15,16,17,18,19,20]. The characteristics of the retrieved patients are shown in Table 3, together with those of the three patients we previously described.

For our analysis, clinical outcomes were classified as follows: first, clinical cure as the complete resolution of clinical signs and symptoms related to the infection or the infection cleared with no positive cultures reported at the end of ceftazidime–avibactam therapy, and second, failure as a lack of clinical response, death due to infection, or recurrent infection. Continuous variables are summarized as medians and interquartile ranges (IQR), whereas categorical variables are presented as absolute and relative frequencies.

Gender was reported in all cases, and ten patients were male (71.4%). The median age was 33 years (IQR 55.5–26.75). Prosthetic joint infections constituted 14.3% (2/14) of the BJIs, with osteomyelitis accounting for 85.7%.

Most infections were monomicrobial (64.3%), with Enterobacteriaceae strains and *P. aeruginosa* accounting for 64.3% (9/14) and 42.8% (6/14), respectively. Five BJIs were polymicrobial, and three of these also included gram-positive cocci (*Enterococcus faecium*, *S. aureus, S. epidermidis*), mycobacteria (*Mycobacterium xenopi*), or fungi (*Rhizopus spp.*).

All BJIs treated with CAZ–AVI involved at least one MDR or XDR gram-negative bacteria, specifically serine carbapenemases producing which was reported in 47% of the cases (KPC and OXA-type β-lactamases in 62.5% and 37.5%, respectively). Four patients (23.5%) presented with BJI due to metallo-β-lactamase producing strains and were treated with aztreonam in conjunction with CAZ–AVI. In five instances (29.4%), although the isolated microorganisms were resistant to carbapenems, the resistance mechanisms were not reported.

CAZ–AVI was prescribed at a daily dose of 2.5 g every eight hours, except for one patient treated with a daily dose of 2.5 g every six hours, for a median of 42 days (IQR 45–24.2), usually on combination therapy in 85.7% of the cases (12/14), particularly with aztreonam and aminoglycosides in 50% (6/12) and 33.3% (4/12) of the cases, respectively.

It is interesting to note that combination antibiotic therapy was used in nearly all cases. The role of combination therapy is still unclear.

Davido et al. demonstrated in an experimental model of KPC-producing *Klebsiella pneumoniae* and OXA-48/ESBL-producing *Escherichia coli* osteomyelitis that the combination of CAZ–AVI with colistin, gentamicin, or fosfomycin, the latter not for KPC-producing *K. pneumoniae*, allowed significant bone sterilization in infected rabbits and was more effective than other single therapy [22,23].

Regarding combination therapy in *P. aeruginosa* infection, Montero et al. performed a time-kill analysis of the effectiveness of CAZ–AVI alone and in combination with other antibiotics (aztreonam, meropenem, colistin, and amikacin) against a collection of XDR-*P. aeruginosa* isolates. In the CAZ–AVI-susceptible isolates, combination therapies with colistin and amikacin achieved a higher overall reduction in bacterial load than other treatments. In the CAZ–AVI-resistant isolates, colistin showed a synergistic or additive effect against all the *P. aeruginosa* isolates and reached bacterial eradication at 4 and 8 h in several bacterial isolates. However, a little regrowth at 24 h, probably for the selection of resistant isolates, was observed [24]. In addition, a recent study found that the combination of CAZ–AVI–fosfomycin was superior to either drug alone in infected patients with high bacterial burdens due to MDR *P. aeruginosa* [25].

Although tolerability to CAZ–AVI beyond 14 days of treatment has not been demonstrated, CAZ–AVI was well tolerated with only one adverse drug reaction, i.e., asymptomatic increase in transaminases, which occurred after fourteen days of treatment in a single patient. In our case (Patient 1), one drug-related side effect, a *C. difficile* infection, was observed after fourteen days of ceftazidime–avibactam therapy.

Regarding the adverse effects of CAZ–AVI, two systematic reviews were conducted. Zhong et al. found no statistically significant differences between CAZ–AVI and comparators in terms of adverse events and serious adverse events [26].

Surprisingly, Sternbach et al., who investigated the safety of CAZ–AVI generally, demonstrated that serious adverse effects are significantly more frequent in CAZ–AVI than in other antimicrobials (mostly carbapenem) without providing more information on the specifics of these effects. *C. difficile*-associated diarrhoea was reported in three trials, with rates of 3/1255 patients in the CAZ–AVI group versus 1/1255 patients in the comparator group [27].

71.4% (10/14) of BJIs had clinical cures, and CAZ–AVI was prescribed for a median of 42 days (IQR 53.5–42). Samples were taken in five of these cases with negative results. A clinical failure occurred in 28.6% (4/14) of the cases, despite two successful microbiological eradication cases (a new pathogen was found at the recurrence). CAZ–AVI was prescribed for a median of 24.5 days (IQR 36–14.7). A recurrence occurred in 21.4% of the cases (3/14). No death occurred in this case series.

Consistent with these reports, a cohort study showed a clinical cure in 77.8% of BJIs (7/9) treated with CAZ–AVI. A clinical failure occurred in two patients despite a microbiological eradication [21].

## 4. Conclusions

In our cases and review, ceftazidime–avibactam demonstrated efficacy and a good safety profile and could be a viable option against BJIs due to MDR or XDR-GNB. However, few studies reported the use of CAZ–AVI for BJI treatment. BJIs have not yet been approved as an indication for CAZ–AVI administration, and more studies are required to define CAZ–AVI PK, the optimal treatment dosages, the safety profile, and the role of combination therapy.

In conclusion, we suggest conducting clinical studies to develop the use of CAZ–AVI in difficult-to-treat gram-negative bacteria BJIs.

## Figures and Tables

**Figure 1 antibiotics-12-01328-f001:**
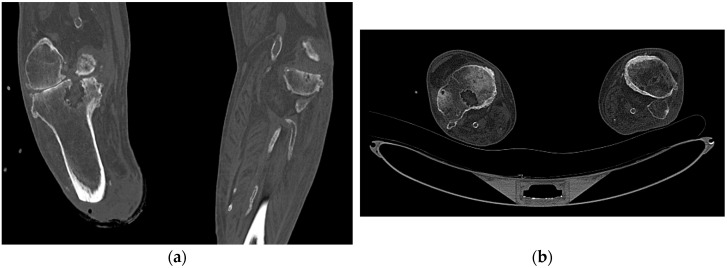
Diagnostic CT scans of Patient 2: coronal (**a**) and axial (**b**) bone window showing a significant area of bone rarefaction at the proximal epiphysis of the right tibia along with interruption of the cortical profile on the posterosuperior side.

**Figure 2 antibiotics-12-01328-f002:**
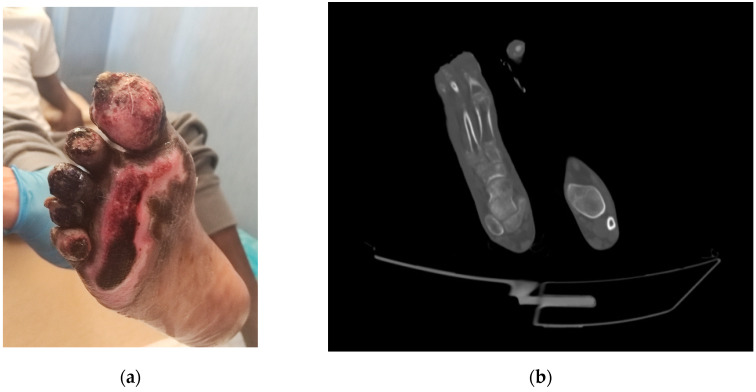
Clinical and radiological appearance of the right foot of Patient 3: (**a**) the wound on admission, with evidence of substance loss in the big toe, second toe, and sole; (**b**) CT scan showing bone rarefaction at the first metatarsal of the right foot.

**Table 1 antibiotics-12-01328-t001:** *Pseudomonas aeruginosa* antibiograms with MIC calculated by Phoenix™ Automated Microbiology System (BD Diagnostics, Sparks, MD, USA).

	Susceptibility Profile (MIC)		
Antimicrobial	Patient 1	Patient 2	Patient 3
Piperacillin–tazobactam	R	R (64/4)	R (64/4)
Ceftazidime	I	R (16)	R (>16)
Ceftazidime–avibactam	S	S (8/4)	S (4/4)
Cefepime	I	R (>8)	R (>8)
Ceftolozane–tazobactam	R	R (>4/4)	R (>4/4)
Imipenem	R	R (>8)	R (>8)
Meropenem	R	R (>16)	R (>16)
Meropenem–vaborbactam	R	R (>8/8)	R (>8/8)
Aztreonam	I	R (>16)	I (8)
Amikacin	R	R (>16)	S (16)
Tobramycin	R	R (>4)	S (≤1)
Ciprofloxacin	R	R (>1)	I (0.25)
Colistin	N/A	S (1)	S (2)

S, susceptible; I, intermediate; R, resistant; N/A, not available.

**Table 2 antibiotics-12-01328-t002:** Laboratory examination at admission.

Laboratory examination	Patient 1	Patient 2	Patient 3
White blood cell (cells/μL)	6400	10,900	5700
Haemoglobin (g/dL)	9.6	12.6	12.1
Platelet (cells/μL)	346,000	217,000	314,000
Serum creatinine (mg/dL)	0.81	0.62	0.64
ALT (U/L)	11	22	12
AST (U/L)	14	15	17
C-reactive protein (mg/L)	10.57	22	0.6

**Table 3 antibiotics-12-01328-t003:** Characteristics of case reports included in the review of the literature and our cases.

			Ceftazidime/Avibactam				Outcome	
First Author and Year	Bacterium	Mechanism of Resistance	Dosage	Antibiotics Combination	Treatment (days)	Surgical Treatment	Clinical Cure	Microbiological Cure
Cani E, 2017 [8]	*K. pneumoniae*	N/A	2.5 g t.i.d.	Amikacin	42	Yes	Yes	N/A
Rico-Nieto A, 2018 [9]	*K. pneumoniae*	OXA-48	2.5 g t.i.d.	Colistin	56	Yes	Yes	N/A
De León-Borrás R, 2018 [10]	*K. pneumoniae*	KPC	N/A	Amikacin, polymyxin B	42	No	Yes	N/A
Rodríguez-Núñez O, 2018 [11]	*P. aeruginosa*	N/A	N/A	Tobramycin, ciprofloxacin	34	Yes	No	Yes
Mittal J, 2018 [12]	*K. pneumoniae*	NDM, OXA-181	2.5 g t.i.d.	Aztreonam	46	Yes	Yes	Yes
Schimmenti A, 2018 [13]	*K. pneumoniae*	KPC	2.5 g t.i.d.	None	14	Yes	Yes	Yes
Alamarat ZI, 2020 [14]	*P. aeruginosa* *K. pneumoniae*	NDM-1 KPC	2.5 g t.i.d.	Aztreonam	14	Yes	No	N/A
Meschiari A, 2021 [15]	*P. aeruginosa* *K. pneumoniae* *P. aeruginosa*	N/A KPC N/A	2.5 g q.i.d. 2.5 g t.i.d.	Aztreonam Aztreonam	42 56	Yes Yes	Yes Yes	N/A N/A
Mularoni A, 2021 [16]	*P. aeruginosa*	VIM-1	2.5 g t.i.d.	Amikacin, aztreonam	42	Yes	Yes	Yes
Ji Z, 2021 [17]	*K. pneumoniae*	N/A	2.5 g t.i.d.	None	21	Yes	Yes	Yes
Eskenazi A, 2022 [18]	*K. pneumoniae*	OXA-48	2.5 g t.i.d.	Moxifloxacin, phage	88	Yes	Yes	Yes
Racenis K, 2022 [19]	*P. aeruginosa*	KPC	2.5 g t.i.d.	Phage	15	Yes	No	No
Rubnitz ZA, 2022 [20]	*C. sedlakii*	NDM	2.5 g t.i.d.	Aztreonam	42	Yes	No	Yes
Present cases, 2023	*P. aeruginosa* *P. aeruginosa* *P. aeruginosa*	N/A N/A N/A	2.5 g t.i.d. 2.5 g t.i.d. 2.5 g t.i.d.	None Fosfomycin Amikacin	42 42 42	No Yes Yes	Yes Yes Yes	N/A N/A N/A

N/A, not available.

## Data Availability

Not applicable.

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
