# Peer review of "Ceftazidime-Avibactam as Osteomyelitis Therapy: A Miniseries and Review of the Literature"

_antibiotics, 2023, doi:10.3390/antibiotics12081328_

Round 1
Reviewer 1 Report
Thanks for the author's submission. In this study, three pseudomonas aeruginosa osteomyelitis cases were treated with Ceftazidime-avibactam monotherapy or combination therapy. There are, however, a few points to remind the authors: 1. The history should be more concise and focused to this infective disease. (eg . Line 82-82 inflammation markers ? Line 65, 91 good conditions? ) 2. Table 1 shows MIC data using which method? Broth dilution method, Agar dilution method, Epsilometer method, or other method? 3. Susceptibility of antibiotics is based on CLSI? Which version? Is there more accurate data? 4. Microorganisms are not written according to international standards. 5. Lin 105, why to combine fosfomycin? 6. Line 20 We think you mean "One patient (33.3%) was diagnosed with Clostridium difficile colitis". "Only one adverse drug reaction" is not suited. 7. The study of Ceftazidime-avibactam and aminoglycoside fractional inhibitory concentration against pseudomonas aeruginosa may be more interesting.
Reviewer 2 Report
In this manuscript, Mancuso et al. describe three cases where patients with P. aeruginosa osteomyelitis were treated with CAZ-AVI, and report the antibiotic susceptibility profiles of the clinical isolates. The authors also provide a comparative review of the existing clinical cases of BJIs treated with CAZ-AVI, finding that the use of CAZ-AVI in combination with other antibiotics allow for high clinical and microbiological cure rates. Overall, this case report is well-written, and will likely be appreciated by the audience of Antibiotics. I only have a few minor cosmetic suggestions below:
1. Lines 68-69: please italicize the word “P. aeruginosa.”
2. Line 145: please consider removing the word “analytically.”
3. Line 157 & 162: please consider changing the percentages to the percentage of the total, rather than the subset of cases.
English language is fine.
Round 2
Reviewer 1 Report
I appreciate your responses, authors.
In patients 2 and 3, MIC was calculated by PhoenixTM automated system. However, what about patient 1?
